# TERRA: EXPLORABLE NATIVE 3D WORLD MODEL WITH POINT LATENTS

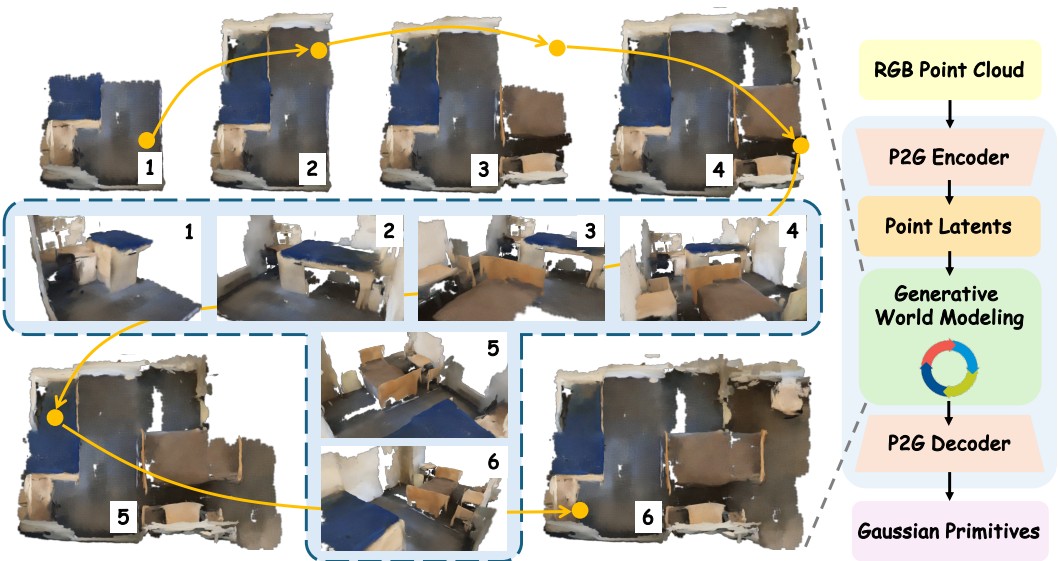

Figure 1: **Method overview.** Unlike conventional world models with pixel-aligned representations, we propose Terra as a native 3D world model that describes and generates 3D environments with point latents. Starting with a glimpse of the environment, Terra progressively explores the unknown regions to produce a coherent and complete world simulation.

## ABSTRACT

World models have garnered increasing attention for comprehensive modeling of the real world. However, most existing methods still rely on pixel-aligned representations as the basis for world evolution, neglecting the inherent 3D nature of the physical world. This could undermine the 3D consistency and diminish the modeling efficiency of world models. In this paper, we present **Terra**, a native 3D world model that represents and generates explorable environments in an intrinsic 3D latent space. Specifically, we propose a novel point-to-Gaussian variational autoencoder (**P2G-VAE**) that encodes 3D inputs into a latent point representation, which is subsequently decoded as 3D Gaussian primitives to jointly model geometry and appearance. We then introduce a sparse point flow matching network (**SPFlow**) for generating the latent point representation, which simultaneously denoises the positions and features of the point latents. Our Terra enables exact multi-view consistency with native 3D representation and architecture, and supports flexible rendering from any viewpoint with only a single generation process. Furthermore, Terra achieves explorable world modeling through progressive generation in the point latent space. We conduct extensive experiments on the challenging indoor scenes from ScanNet v2. Terra achieves state-of-the-art performance in both reconstruction and generation with high 3D consistency.

## 1 INTRODUCTION

World models have emerged as a promising research direction, with the aim of understanding and simulating the underlying mechanics of the physical world (Ha & Schmidhuber, 2018). Unlike Large Language Models (LLMs), which are confined to textual processing (Vaswani et al., 2017;

Brown et al., 2020), world models integrate multimodal visual data to construct a comprehensive and internal representation of the environment (Ha & Schmidhuber, 2018). From learning the evolution of the real world, world models enable various downstream applications, including perception (Min et al., 2024; Lai et al., 2025), prediction (Zheng et al., 2024a; Xiang et al., 2024; Team et al., 2025; Agarwal et al., 2025), reasoning (Bruce et al., 2024; Assran et al., 2023; Huang et al., 2024), and planning (Ren et al., 2025; Assran et al., 2025; Zheng et al., 2024b).

Scene representation is fundamental to world models (Wang et al., 2024c; Team et al., 2025; Zheng et al., 2024a), forming the basis for world evolution. Conventional methods typically rely on 2D image or video representations, simulating world dynamics through video prediction (Agarwal et al., 2025; Bruce et al., 2024; Xiang et al., 2024; Assran et al., 2025). However, the generated videos often lack consistency across frames (Wang et al., 2024c; Huang et al., 2024; Zheng et al., 2024b), as the models do not consider explicit 3D priors and instead learn only implicit 3D cues from the training videos. To address this limitation, a line of work simultaneously predicts RGB images and depth maps to construct a pixel-aligned 2.5D representation (Team et al., 2025; Yang et al., 2025a; Lu et al., 2025). While they integrate geometric constraints into the generation process, learning the multi-view pixel correspondence remains challenging due to the ambiguity of relative camera poses. The physical world is inherently three-dimensional, including objects and their interactions. However, the rendering process only produces a partial 2D observation of the underlying 3D environment, inevitably losing crucial depth and pose information (Mildenhall et al., 2021; Kerbl et al., 2023). This poses critical challenges on the multi-view consistency of world models based on pixel-aligned representations (Lu et al., 2025; Team et al., 2025; Yang et al., 2025a; Wang et al., 2024c).

To address this, we present Terra, a native 3D world model that describes and generates explorable environments with an intrinsic 3D representation, as shown in Figure 1. At its core, we learn a native point latent space that employs spatially sparse but semantically compact point latents as the basis for reconstruction and generation. Accordingly, Terra completely discards pixel-aligned designs and directly learns the distribution of 3D scenes in its most natural form, achieving 3D consistency without bells and whistles. To elaborate, we propose a novel point-to-Gaussian variational autoencoder (P2G-VAE) that converts 3D input into the latent point representation. The asymmetric decoder subsequently maps these point latents to rendering-compatible 3D Gaussian primitives to jointly model geometry and appearance. The P2G-VAE effectively reduces the redundancy in the input 3D data and derives a compact latent space suitable for generative modeling. Furthermore, we propose a sparse point flow matching model (SPFlow) to learn the transport trajectory from the noise distribution to the target point distribution. The SPFlow simultaneously denoises the positions and features of the point latents to leverage the complementary nature of geometric and textural attributes to foster their mutual enhancement. Based on P2G-VAE and SPFlow, we formulate the explorable world model as an outpainting task in the point latent space, which we approach through progressive training with three stages: reconstruction, unconditional generative pretrain, and masked conditional generation. We conduct extensive experiments on the challenging indoor scenes from ScanNet v2 (Dai et al., 2017). Our Terra achieves state-of-the-art performance in both reconstruction and generation with high 3D consistency and efficiency.

## 2 RELATED WORK

**2D world models.** Early attempts in world models focus on image or video representations, thanks to the exceptional performance of 2D diffusion models (Ho et al., 2020; Song et al., 2020; Rombach et al., 2022; Blattmann et al., 2023; Peebles & Xie, 2023). DriveDreamer (Wang et al., 2024c) and Sora (OpenAI, 2024) represent pioneering image-based and video-based world models, respectively, both leveraging diffusion models to achieve view-consistent and temporally coherent world modeling. Subsequent research efforts focus primarily on enhancing the temporal consistency (Henschel et al., 2025; Huang et al., 2024; Yin et al., 2023), spatial coherence (Yu et al., 2024b; Wu et al., 2025; Chen et al., 2025a), physical plausibility (Assran et al., 2023; Agarwal et al., 2025; Assran et al., 2025), and interactivity (Xiang et al., 2024; He et al., 2025; Wang et al., 2025b) of generated videos. Several studies also explore integrating the language modality with conventional methods to train multimodal world models (Zheng et al., 2024b; Kondratyuk et al., 2023). Recently, Genie-3 (Bruce et al., 2024) has emerged as one of the most successful video world models, which enables excellent photorealism, flexible interaction, and real-time generation. Despite the promising advancements, 2D world models learn the evolution of the real world solely from image or video data, overlooking

the inherent 3D nature of the physical environments. This lack of sufficient 3D priors often results in failures to maintain 3D consistency in the generated outputs. Moreover, 2D world models require multiple generation passes to produce results with different viewing trajectories.

**2.5D world models.** To incorporate explicit 3D clues into world models, and also leverage the generative prior from 2D diffusion networks, a line of work (Hu et al., 2025; Gu et al., 2025; Chen et al., 2025b; Yang et al., 2025b; Huang et al., 2025; Yu et al., 2025) proposes to jointly predict depth and RGB images as a pixel-aligned 2.5D representation. ViewCrafter (Yu et al., 2024b) employs an off-the-shelf visual geometry estimator (Wang et al., 2024a) to perform depth and pose prediction, which is then used in novel view reprojection. Prometheus (Yang et al., 2025a) trains a dual-modal diffusion network for joint generation of depth and RGB images conditioned on camera poses. Furthermore, several work (Team et al., 2025; Lu et al., 2025) converts camera poses to 2D Plücker coordinates, in order to consider camera poses, depth and RGB images in a unified framework. In general, these methods try to learn the joint distribution of depth, poses and texture to improve 3D consistency. However, these factors are deeply coupled with each other by the delicate perspective transformation, which is often challenging for neural networks to learn in an implicit data-driven manner. We propose a native 3D world model that represents and generates explorable environments with a native 3D latent space, and guarantees multi-view consistency with 3D-to-2D rasterization.

**Native 3D generative models.** Most relevant to our work are native 3D generative models that also employ 3D representations. Pioneering work in this field focuses on point cloud generation. Luo & Hu (2021) proposes the first diffusion probabilistic model for 3D point cloud generation. Vahdat et al. (2022) later extend this paradigm to support latent point diffusion, followed by advancements in architecture (Ren et al., 2024b), frequency analysis (Zhou et al., 2024) and flow matching (Vogel et al., 2024; Hui et al., 2025). However, these methods are confined to object or shape level generation and are unable to synthesize textured results, which greatly restricts their application. To integrate texture, Lan et al. (2025) and Xiang et al. (2025) adopt Gaussian splatting as the 3D representation, but they are still limited to object generation. Zheng et al. (2024a) and Ren et al. (2024a) extend to scene-level 3D occupancy generation, but the occupancy is coarse in granularity and does not support rendering applications. In summary, existing methods are restricted to either object-level fine-grained or scene-level coarse geometry generation. In contrast, we construct the first native 3D world model with both large-scale and rendering-compatible 3D Gaussian generation.

## 3 PROPOSED APPROACH

### 3.1 LATENT POINT REPRESENTATION

We present Terra as a native 3D world model that represents and generates explorable environments with an intrinsic 3D representation. Figure 2 outlines the overall pipeline. Formally, we formulate explorable world models as first generating an initial scene $S_0$ and progressively expanding the known regions to produce a coherent and infinite world simulation $\mathbb{S}$:

$$S_0 = g(\emptyset, C_0; \theta), \quad S_i = g(\mathbb{S}_{i-1}, C_i; \theta), \quad \mathbb{S}_i = \{S_0, S_1, ..., S_i\}, \tag{1}$$

where subscripts denote exploration steps, and $g(\mathbb{S}_{i-1}, C_i; \theta)$ represents the model with learnable parameters $\theta$ that generates the next-step exploration result $S_i$ based on the set of previously known regions $\mathbb{S}_{i-1}$ and the current conditional signal $C_i$. Conventional world models with pixel-aligned representations instantiate $S_i$ with colors $R$, depths $D$ and poses $T$ from different viewpoints:

$$S_i = [(R_i^{(n)}, D_i^{(n)}, T_i^{(n)})|_{n=1}^N], \tag{2}$$

where $N$ denotes the number of views in a single generation step and the superscript $(n)$ is the view index. On the other hand, multi-view consistency for Lambert's model can be formulated as:

$$R^{(n)}|_{\boldsymbol{x}^{(n)}} = R^{(m)}|_{\boldsymbol{x}^{(m)}}, \quad d^{(n)}\boldsymbol{x}^{(n)} = T^{(n)}\boldsymbol{x}, \quad n, m = 1, 2, ..., N, \tag{3}$$

where $\boldsymbol{x}, \boldsymbol{x}^{(n)}, d^{(n)}, R^{(n)}|_{\boldsymbol{x}^{(n)}}$ denote the 3D coordinates of a visible point, the image coordinates of $\boldsymbol{x}$ in the $n$-th view, the depth of $\boldsymbol{x}$ in the $n$-th view, and sampling $R^{(n)}$ at $\boldsymbol{x}^{(n)}$, respectively. Eq. (3) requires that different pixels on separate views should share the same color if they are the projections of the same visible 3D point. Therefore, the ideal representation for conventional world models should be the combination of Eq. (2) and (3), i.e. the multi-view colors, depths and poses satisfying

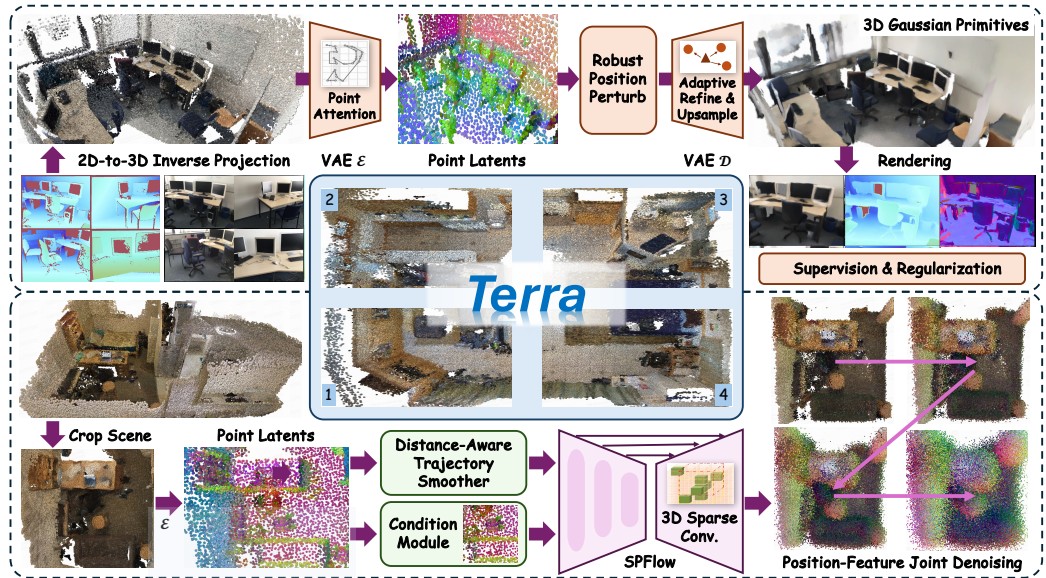

Figure 2: **Overall pipeline.** Terra consists of a point-to-Gaussian VAE and a sparse point flow matching model. The P2G-VAE effectively learns the transformation from input RGB point cloud to point latents, and then to 3D Gaussian primitives. The SPFlow learns the joint distribution of geometry and appearance. Both P2G-VAE and SPFlow adopt native sparse 3D architectures.

the reprojection constraint. Unfortunately, it is often challenging for neural networks to learn this constraint in an implicit data-driven manner, leading to multi-view inconsistency. ViewCrafter (Yu et al., 2024b) bypasses this problem by taking smaller steps ($N = 1$) in every generation and explicitly projects previous contexts onto the novel view to enforce reprojection consistency. While effective, this approach significantly compromises the efficiency of exploration.

Different from the pixel-aligned counterparts, we propose latent point representation $P$ as a native 3D descriptor of the environment: $S_i = P_i \in \mathbb{R}^{M_i \times (3+D)}$, where $M_i$ and $3 + D$ denote the number of point latents for the $i$-th exploration step and the sum of the dimensions for 3D coordinates and features, respectively. The latent point representation is similar to the actual point cloud, located sparsely on the surface of objects, but limited in number and with semantically meaningful latent features. It also supports adapting $M_i$ according to the complexity of different regions and integrating historical contexts by simply concatenating previous $P_i$s. This design completely discards the view-dependent elements (Eq. (2)) and the reprojection constraint (Eq. (3)) from the exploration process and instead models the environment with 3D points $x$ directly. These point latents can be transformed into 3D Gaussian primitives for rasterization, naturally satisfying 3D consistency and enabling flexible rendering from any viewpoint without rerunning the generation pipeline.

## 3.2 POINT-TO-GAUSSIAN VARIATIONAL AUTOENCODER

We design the P2G-VAE to effectively generate the latent point representation from the input scene and decode it into 3D Gaussian primitives. We suppose the input scene is described by a colored point cloud $Q \in \mathbb{R}^{B \times 6}$ to provide necessary 3D information, where $B$ and 6 represent the number of points and the sum of dimensions for 3D coordinates and color, respectively. We build our P2G-VAE based on the point transformer architecture (Zhao et al., 2021) for efficiency. Apart from removing the residual connections in the original PTv3 (Wu et al., 2024a), we include the following novel designs for a robust latent space and effective Gaussian decoding, as shown in Figure 3.

**Robust position perturbation.** In conventional VAEs (Kingma & Welling, 2014), it is common to regularize the latent features with a Kullback-Leibler Divergence loss $L_{KL}$ to align the feature distribution with a standard normal distribution. However, it is nontrivial to generalize this practice to unstructured point latents where 3D coordinates themselves contain crucial geometry information. Directly regularizing the coordinates to approximate Gaussian noise would have an adverse effect on the locality of point latents and the associated local structures. To this end, we propose a robust position perturbation technique which perturbs the coordinates of point latents with a predefined

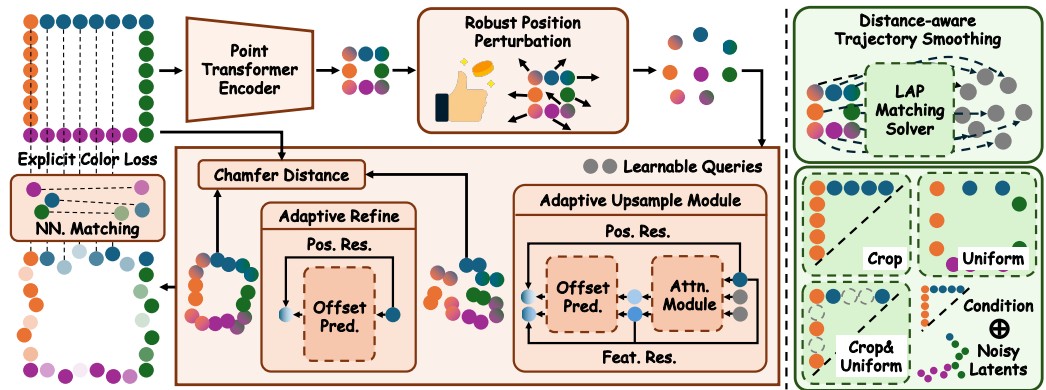

Figure 3: **Method details.** LAP, Pos. Res., Feat. Res. and NN. denote linear assignment problem, position residual, feature residual and nearest neighbor, respectively.

Gaussian noise $\boldsymbol{n} \sim \mathcal{N}(\boldsymbol{0}, \sigma^2 \boldsymbol{I}_3)$ where $\sigma$ is a hyperparameter for noise intensity:

$$\boldsymbol{P} = [(\boldsymbol{p}^{(m)} \in \mathbb{R}^3, \boldsymbol{f}^{(m)} \in \mathbb{R}^D)|_{m=1}^M], \quad \boldsymbol{p} = \hat{\boldsymbol{p}} + \boldsymbol{n}, \quad \boldsymbol{f} \sim \mathcal{N}(mean(\hat{\boldsymbol{f}}), \mathrm{diag}(var(\hat{\boldsymbol{f}}))), \quad (4)$$

where we split point latents $\boldsymbol{P}$ into $M$ position-feature pairs $(\boldsymbol{p}, \boldsymbol{f})$ and omit the exploration step for simplicity. The $\hat{\boldsymbol{p}}$ and $\hat{\boldsymbol{f}}$ denote the positions and features of points as input to the VAE bottleneck, and $mean(\cdot)$, $var(\cdot)$ are the functions to calculate the mean and variance of the latent features $\boldsymbol{f}$. The robust position perturbation enhances the robustness of the VAE decoder against slight perturbations over the positions of point latents. Further, it greatly improves the generation quality since generated samples inevitably contain a certain level of noise, similar to our perturbation process.

**Adaptive upsampling and refinement.** Given point latents after downsampling and perturbation, the VAE decoder should upsample them to an appropriate number and restore the dense structure. To achieve this, we introduce the adaptive upsampling and refinement modules. The adaptive upsampling module splits each point $(\boldsymbol{p}, \boldsymbol{f})$ into $K$ child points $(\boldsymbol{p}^{(k)}, \boldsymbol{f}^{(k)})|_{k=1}^K$ with $K$ learnable queries $\boldsymbol{q}^{(k)}|_{k=1}^K$. These queries first interact with each point for contexts, and then each query predicts a relative displacement $disp(\cdot)$ and a residual feature $resf(\cdot)$ for the corresponding child point:

$$\hat{\boldsymbol{q}}^{(k)}|_{k=1}^K = ups(\boldsymbol{f}, \boldsymbol{q}^{(k)}|_{k=1}^K), \quad \boldsymbol{p}^{(k)} = \boldsymbol{p} + disp(\hat{\boldsymbol{q}}^{(k)}), \quad \boldsymbol{f}^{(k)} = \boldsymbol{f} + resf(\hat{\boldsymbol{q}}^{(k)}), \quad (5)$$

where $ups(\cdot)$ denotes the point-query interaction module. This design enables controllable upsampling and avoids the complex mask-guided trimming operation in conventional methods (Ren et al., 2024a). Similar to the upsampling module, the adaptive refinement module further adjusts the point positions with offsets predicted from the point features: $\boldsymbol{p}' = \boldsymbol{p} + refine(\boldsymbol{f})$. These two modules progressively densify and refine the point positions, restoring a dense and meaningful structure.

**Comprehensive regularizations.** To supervise the output Gaussian primitives, we employ the conventional rendering supervisions including L2, SSIM and LPIPS (Zhang et al., 2018) losses. In addition, we also incorporate other losses to improve the reconstructed geometry and regularize the properties of Gaussians for better visual quality. 1) We optimize the chamfer distances $L_{cham}$ between the input point cloud and intermediate point clouds as the outputs of upsampling and refinement modules, which provides explicit guidance for the prediction of position offsets. 2) We use the normal $L_{norm}$ and effective rank $L_{rank}$ (Hyung et al., 2024) regularizations to regularize the rotation and scale properties of Gaussians. 3) We propose a novel explicit color supervision $L_{color}$, which directly aligns the color of each Gaussian with the color of the nearest point in the input point cloud. This loss bypasses the rasterization process and thus is more friendly for optimization. The overall loss function for our P2G-VAE can be formulated as:

$$L_{vae} = L_{l2} + \lambda_1 L_{ssim} + \lambda_2 L_{lpips} + \lambda_3 L_{cham} + \lambda_4 L_{norm} + \lambda_5 L_{rank} + \lambda_6 L_{color} + \lambda_7 L_{kl}. \quad (6)$$

## 3.3 NATIVE 3D GENERATIVE MODELING

We use flow matching (Lipman et al., 2022) for generative modeling of the latent point representation. Formally, we gradually add noise $\boldsymbol{N} \sim \mathcal{N}(\boldsymbol{0}, \boldsymbol{I})$ to both the positions and features of point latents $\boldsymbol{P} \in \mathbb{R}^{M \times (3+D)}$ with a schedule $t \in [0, 1]$ in the diffusion process, and predict the velocity

Table 1: **Reconstruction performance.** RGB PC. and Rep. Range represent colored point cloud and representation range of the output Gaussian, respectively. We select 20 random scenes from the validation set to reconstruct offline Gaussians as input to Can3Tok*.

| Method | Input Type | Rep. Range | PSNR↑ | SSIM↑ | LPIPS↓ | Abs. Rel.↓ | RMSE↓ | $\delta 1$↑ |
|---|---|---|---|---|---|---|---|---|
| PixelSplat | RGB | Partial | 18.165 | 0.686 | 0.493 | 0.094 | 0.287 | 0.832 |
| MVSplat | RGB | Partial | 17.126 | 0.621 | 0.552 | 0.139 | 0.326 | 0.824 |
| Prometheus | RGBD | Partial | 17.279 | 0.644 | **0.448** | 0.087 | 0.251 | 0.901 |
| Can3Tok* | Gaussian | Complete | 19.578 | 0.733 | 0.514 | 0.031 | 0.151 | 0.973 |
| **Terra** | RGB PC. | Complete | **19.742** | **0.753** | 0.530 | **0.026** | **0.137** | **0.978** |

vector $\boldsymbol{V} \in \mathbb{R}^{M \times (3+D)}$ given noisy latents $\boldsymbol{P}_t$ in the reverse process:

$$\boldsymbol{P}_t = t\boldsymbol{P} + (1-t)\boldsymbol{N}, \quad \boldsymbol{V} = \mathcal{F}(\boldsymbol{P}_t, t; \boldsymbol{\phi}), \tag{7}$$

where $\mathcal{F}(\cdot, \cdot; \boldsymbol{\phi})$ denotes a UNet (Peng et al., 2024) with learnable parameters $\boldsymbol{\phi}$ based on 3D sparse convolution. The training objective can now be formulated as:

$$L_{flow} = \mathbb{E}_{t \sim \mathcal{U}[0,1], \boldsymbol{P} \sim \mathcal{P}, \boldsymbol{N} \sim \mathcal{N}(\boldsymbol{0}, \boldsymbol{I})} ||\mathcal{F}(\boldsymbol{P}_t, t; \boldsymbol{\phi}) - (\boldsymbol{P} - \boldsymbol{N})||^2, \tag{8}$$

where $\mathcal{P}$ denotes the ground truth distribution of point latents $\boldsymbol{P}$. During inference, we start from sampled Gaussian noise and progressively approach clean point latents along the trajectory determined by the predicted velocity vector. Note that we simultaneously diffuse the positions and features to learn the joint distribution of geometry and texture and facilitate their mutual enhancement.

**Distance-aware trajectory smoothing.** Conventional flow matching applied to grid-based latents naturally matches noises and latents according to their grid indices. However, it would complicate the velocity field and the denoising trajectory if we simply match the point positions with noise samples based on their indices in the sequence (Hui et al., 2025). Intuitively, it is unreasonable to denoise a leftmost noise sample to a rightmost point. To address this, we propose a distance-aware trajectory smoothing technique that effectively straightens the transport trajectory and facilitates convergence for unstructured point flow matching. Since it is more reasonable to choose a closer noise sample as the diffusion target than a farther one, we optimize the matching $\mathcal{M}$ between point positions and noise samples to minimize the sum of distances between point-noise pairs:

$$\mathcal{M}^* = \text{argmin}_{\mathcal{M}} \sum_{m=1}^{M} ||\boldsymbol{p}^{(m)} - \boldsymbol{N}_{\mathcal{M}_m, :3}||^2, \quad \mathcal{M} = \text{reorder}([1, 2, ..., M]), \tag{9}$$

where $\boldsymbol{N}_{\mathcal{M}_m, :3}$ denotes the position of the noise sample assigned to the $m$-th point latent. We apply the Jonker-Volgenant algorithm (Jonker & Volgenant, 1987) to efficiency solve Eq. (9).

**Simple conditioning mechanism.** For an explorable model, we employ multi-stage training that consists of reconstruction, unconditional generative pretraining, and masked conditional generation. For masked conditions, we introduce three types of conditions to support different exploration styles: cropping, uniform sampling, and their combinations. We randomly crop a connected 3D region from the point latents as the cropping condition to unlock the ability to imagine and populate unknown regions. We uniformly sample some of the point latents across the scene as the uniform sampling condition to enable the model to refine known regions. We also use their combinations and first crop a connected 3D region and then apply uniform sampling inside it to simulate RGBD conditions. We concatenate the conditional point latents with the noisy ones and fix the condition across the diffusion process to inject conditional guidance even at the early denoising stage.

## 4 EXPERIMENTS

### 4.1 DATASETS AND METRICS

We conduct extensive experiments on the challenging indoor scenes from the ScanNet v2 (Dai et al., 2017) dataset, which is widely adopted in embodied perception (Yu et al., 2024a; Wu et al., 2024b) and visual reconstruction (Wang et al., 2024a; 2025a). The dataset consists of 1513 scenes in total, covering diverse room types and layouts. Each scene is recorded by an RGBD video with semantic and pose annotations for each frame. We unproject the color and depth maps into 3D space using the poses to produce colored point clouds as input to our P2G-VAE. In the generative training,

Figure 4: **Visualization for reconstruction.** Terra achieves photorealistic rendering quality for RGB and depth, and learns to complete the partial objects caused by the sensor failure in dark regions.

Table 2: **Generation Performance.** CD and EMD denote Chamfer and earth mover's distances, respectively. Terra achieves exceptional geometry generation quality compared with other methods.

| Method | Repr. | Unconditional | | | | Image Conditioned | | | |
|---|---|---|---|---|---|---|---|---|---|
| | | P-FID↓ | P-KID(%)↓ | FID↓ | KID(%)↓ | CD↓ | EMD↓ | FID↓ | KID(%)↓ |
| Prometheus | RGBD | 32.35 | 12.481 | **263.3** | **10.726** | 0.374 | 0.531 | **208.3** | **12.387** |
| Trellis | 3D Grid | 19.62 | 7.658 | 361.4 | 23.748 | 0.405 | 0.589 | 314.9 | 24.713 |
| **Terra** | Point | **8.79** | **1.745** | 307.2 | 18.919 | **0.217** | **0.474** | 262.4 | 20.283 |

we preprocess the point latents from the VAE encoder by randomly cropping a smaller rectangular region in the x-y plane and filtering out overly sparse and noisy samples. We follow Wang et al. (2024b) and split the dataset into 958 and 243 scenes for training and validation, respectively.

We evaluate Terra on the reconstruction, unconditional, and image-conditioned generation tasks. For reconstruction, we compare Terra with three lines of methods: PixelSplat (Charatan et al., 2024) and MVSplat (Chen et al., 2024) with RGB input, Prometheus (Yang et al., 2025a) with RGBD input, and Can3Tok (Gao et al., 2025) with offline reconstructed Gaussians as input. We use PSNR, SSIM, and LPIPS metrics for visual quality, and Abs. Rel., RMSE, and $\delta 1$ metrics for depth accuracy. For generative tasks, we compare Terra with Prometheus using RGBD (Yang et al., 2025a) representation, and Trellis (Xiang et al., 2025) using 3D grid representation. We retrain these baselines on ScanNet v2 using their official code for a fair comparison. For unconditional generation, we adopt point cloud FID (P-FID) and point cloud KID (P-KID) for geometry quality, and FID and KID for visual quality. Regarding image-conditioned generation, we adopt the Chamfer distance and earth mover's distance for geometry quality, and FID and KID for visual quality.

## 4.2 IMPLEMENTATION DETAILS

We construct the P2G-VAE based on PTv3 (Wu et al., 2024a), removing all residual connections and integrating the designs proposed in Section 3.2. We perform downsampling with stride 2 for 3 times in the encoder, reducing the number of points from 1 million to around 5000. In the decoder, we upsample the points also for 3 times with $K = 7, 3, 3$, respectively. We train the P2G-VAE for 36K iterations with an AdamW (Loshchilov & Hutter, 2017) optimizer. Regarding the SPFlow, we employ the OA-CNNs (Peng et al., 2024) as the UNet backbone. We crop a random region with a size of $2.4 \times 2.4$ m$^2$ from a complete scene as the input sample. We train the SPFlow for 100K and 40K iterations for unconditional pretrain and conditional generation, respectively.

## 4.3 MAIN RESULTS

**Reconstruction.** We report the results in Table 1. PixelSplat (Charatan et al., 2024) and MVSplat (Chen et al., 2024) do not include 3D geometry information as input, and thus they might perform worse compared with others using depth or Gaussian input. Prometheus (Yang et al., 2025a) achieves the best LPIPS because it is pretrained with a 2D diffusion model, which excels

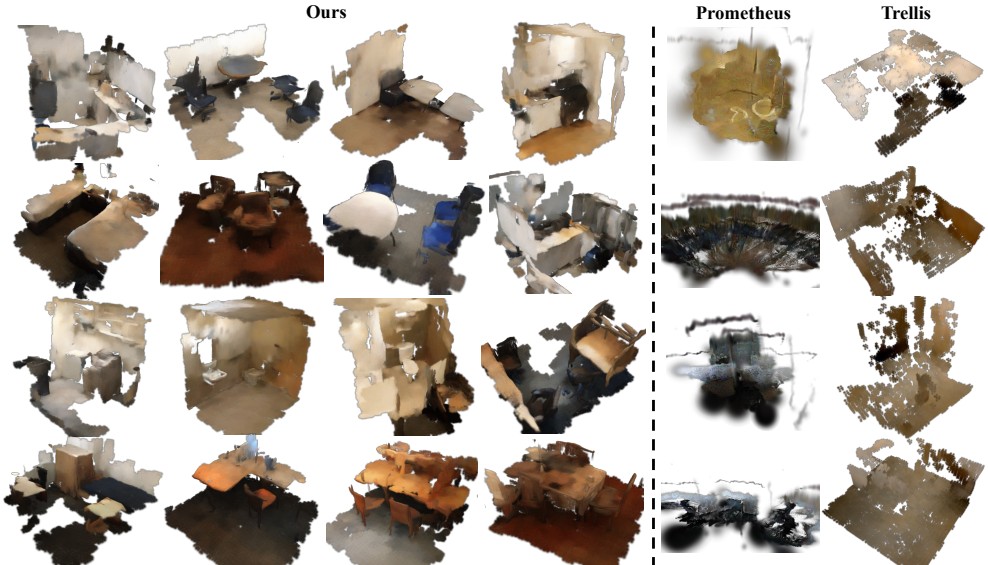

Figure 5: **Visualization for unconditional generation.** Only Terra is able to generate diverse and reasonable scenes while Prometheus and Trellis lack consistent geometry and texture, respectively.

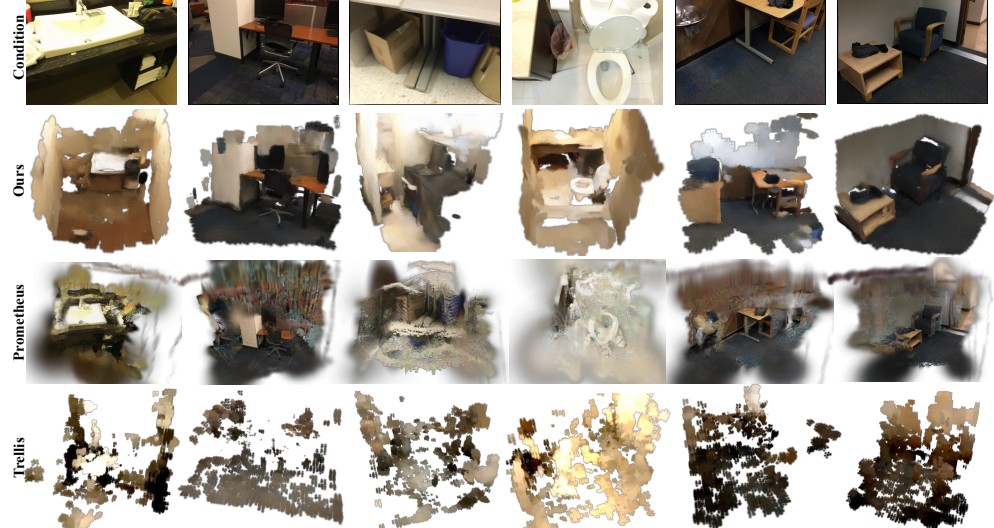

Figure 6: **Visualization for image conditioned generation.** Both Terra and Prometheus are able to produce plausible images while the geometry consistency is far better than Prometheus.

at image quality. Our Terra achieves the best results for all metrics except LPIPS, even better than Can3Tok (Gao et al., 2025) using offline reconstructed Gaussians, demonstrating the effectiveness of our P2G-VAE. Furthermore, Terra is able to reconstruct the whole scene in a single forward pass and also complete partial objects with incorrect depth measurement as shown by Figure 4.

**Unconditional generation.** We report the results in Table 2. Our method achieves better P-FID and P-KID than Prometheus with 2.5D representation and Trellis (Xiang et al., 2025) with 3D grid representation, validating the superiority of point latents in modeling the geometry distribution. However, Terra performs worse compared to Prometheus in image quality metrics FID and KID, because Prometheus, with 2D diffusion pretrain, is able to synthesize plausible images even though the underlying 3D structures could be corrupted. We provide visualization results in Figure 5, where only Terra generates both reasonable and diverse 3D scenes while the results of other methods either lack accurate 3D structure or vivid textures.

**Image conditioned generation.** We report the results in Table 2 and Figure 6. Given a conditional image, we first unproject it into 3D space with depth and intrinsics to produce a colored point cloud. Then we can formulate the image conditioned generation task as outpainting in the point latent space. Our Terra achieves better performance in chamfer distance and earth mover's distance,

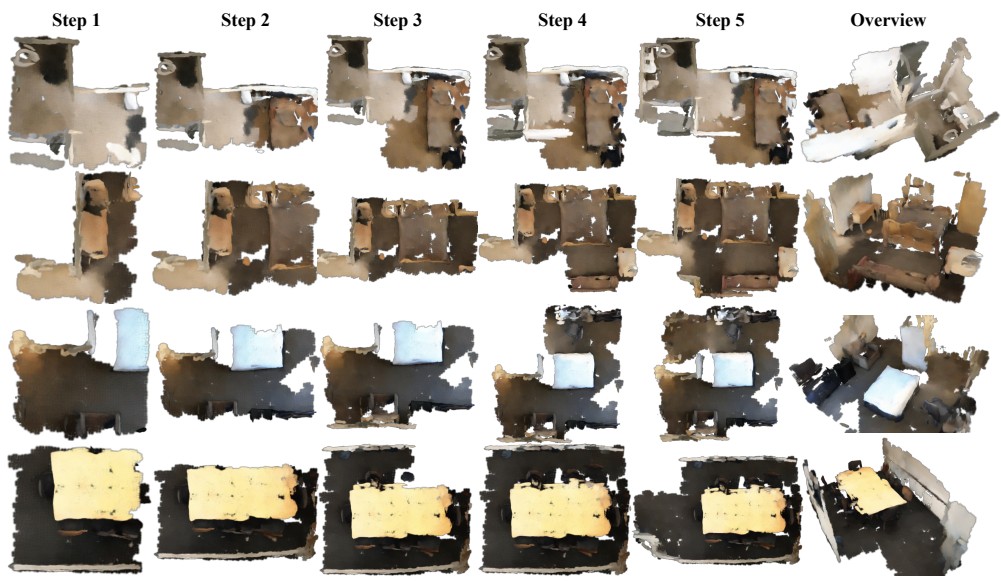

Figure 7: **Visualization for explorable world model.** Terra is able to generate both coherent and diverse room layouts with plausible textures from step-by-step exploration.

Table 3: **Ablation Study** to validate the effectiveness of our design choices.

| Method | Reconstruction | | | | Unconditional Generation | | | |
|---|---|---|---|---|---|---|---|---|
| | PSNR↑ | SSIM↑ | Abs. Rel.↓ | RMSE↓ | P-FID↓ | P-KID(%)↓ | FID↓ | KID(%)↓ |
| w.o. Robust Position Perturbation | **20.487** | **0.783** | **0.023** | **0.132** | 15.28 | 5.218 | 349.3 | 21.884 |
| w.o. Adaptive Upsampling and Refine | 18.749 | 0.711 | 0.042 | 0.157 | 12.48 | 4.764 | 341.8 | 21.760 |
| w.o. Explicit Color Supervision | 19.582 | 0.739 | 0.030 | 0.144 | 10.61 | 3.142 | 327.9 | 19.418 |
| w.o. Dist.-aware Trajectory Smoothing | 19.742 | 0.753 | 0.026 | 0.137 | 24.84 | 11.387 | 401.8 | 27.482 |
| **Terra** | 19.742 | 0.753 | 0.026 | 0.137 | **8.79** | **1.745** | **307.2** | **18.919** |

demonstrating better geometry quality. Prometheus still achieves better FID and KID even though the visualizations show evident multi-view inconsistency.

**Explorable world model.** We visualize the results for explorable world model in Figure 7. We start from a single step generation, and progressively extend the boundary to explore the unknown regions. In each step, we choose a random direction for exploration, take a step forward, and generate the next-step result with part of the known regions as condition. Our Terra is able to synthesize both coherent and diverse room layouts with plausible textures, validating the effectiveness of Terra.

## 4.4 ABLATION STUDY

We conduct comprehensive ablation study to validate the effectiveness of our designs in Table 3. Although position perturbation for point latents degrades reconstruction performance, it is crucial for the generative training because it significantly improves the robustness of the VAE decoder against positional noise. Both adaptive upsampling and refinement and explicit color supervision enhance the reconstruction performance and also the generation quality. Distance-aware trajectory smoothing takes effect in the generative training and is critical for the convergence of the model.

## 5 CONCLUSION

In this paper, we propose Terra as a native 3D world model that describes and generates explorable 3D environments with point latents. The point latents naturally satisfy the 3D consistency constraint crucial to world models as a native 3D representation, and support flexible rendering from any given viewpoint with a single generation process. To learn the intrinsic distribution of 3D data with point latents, we design the P2G-VAE and SPFlow networks for dimensionality reduction and generative modeling, respectively. We conduct experiments on ScanNet v2 with reconstruction, unconditional generation and image conditioned generation tasks, and Terra achieves the best overall performance both quantitatively and qualitatively. Furthermore, Terra is able to explore the unknown regions in a progressive manner and produce a large-scale and coherent world simulation.

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
