# OpenReview forum: "Terra: Explorable Native 3D World Model with Point Latents"
_ICLR.cc/2026/Conference — ICLR 2026 Conference Withdrawn Submission_

### Official Review · Reviewer_FWFD · 2025-10-27

**Soundness:** 3
**Presentation:** 3
**Contribution:** 3
**Rating:** 4
**Confidence:** 4

**Summary:**

This paper introduces Terra, a native 3D world model designed to represent, generate, and progressively explore 3D environments. The authors argue that conventional world models, which rely on 2D pixel-aligned representations, struggle with 3D consistency and modeling efficiency. Terra addresses this by operating directly in an intrinsic 3D latent space using point latents. Terra proposes a Point-to-Gaussian Variational Autoencoder (P2G-VAE) that transforms colored 3D point cloud to 3D Gaussians, and proposes a Sparse Point Flow matching network (SPFlow) to learn the latent point distribution. Authors show Terra capabilities to reconstruction the scene, do uncondition generation and image-conditioned generation.

**Strengths:**

1. Authors propose several novel techniques to facilitate the VAE and flow-matching learning, like Robust position perturbation, Adaptive upsampling and refinement, etc.
2. The final 3D Gaussian representation naturally supports multi-view consistency
3. The model can progressively generate a large-scale, coherent world simulation step-by-step.

**Weaknesses:**

1. The accuracy and completeness of the input point cloud significantly affect the model performance, no matter in the reconstruction (point to Gaussian) task or the generation task. As shown in Figure 4 and Figure 5, even Terra can learn to complete the partial objects caused by the sensor failure in dark regions, the output Gaussians still have holes.
2. Continuing from the previous one, in your image-conditioned generation, the accuracy of depth estimation may directly affect the quality of the generation. Once the depth estimator fails or the input image is out of domain, the model might fail as well.
3. Another baseline can be SCube[1], which uses voxels instead of points as an intermediate representation, decoding per-voxel Gaussians for rendering. It would be interesting to see their comparison or theoretical analysis.

[1] SCube: Instant Large-Scale Scene Reconstruction using VoxSplats, NeurIPS 2024

**Questions:**

1. Can the author elaborate on the scalability of this method? 3D data is not easily obtainable and may contain noise. Yet this method strongly relies on 3D input data.
2. training time is not reported.
3. What is the maximum range of generation supported in a single inference and step-by-step exploration?
4. In 4.3 main results - Reconstruction, are you reporting the metrics on novel views or just input views?

---

> ### Author Response · Authors · 2025-11-14
>
> We thank the reviewer for the constructive comments. We provide our answers below.
> 1. **[W1]** Quality of input 3D data.
>
> Thanks for the nice suggestion. Indeed, the performance of Terra could be affected by the quality of the training 3D data, just like any other deep learning method. The fused point cloud from ScanNet v2 often contains noisy points, random holes and occluded parts due to sensor failure and limited viewpoints, which confuses Terra in learning the 3D geometry distribution. In addition, the images from ScanNet v2 suffer from serious motion blur which makes it hard to learn a high-fidelity texture. Despite this, we are actively collecting more high-quality 3D data to train our model.
>
> 2. **[W2]** Reliance on depth.
>
> Thanks for the nice advice. The image-conditioned generation of Terra indeed requires GT or estimated depth and the accuracy of depth could affect the performance of Terra. One way to improve the robustness of Terra might be treating the partial point cloud from the conditioning image as a noisy one and also denoising it in the generation process instead of completely keeping it the same. We will provide more analysis in the revised version.
>
> 3. **[W3]** Comparison with SCube.
>
> Thanks for the nice suggestion. SCube is a representative 3D reconstruction / generation method with 3D occupancy representation. It is actually very similar to Trellis, which also uses 3D occupancy representation and a geometry-texture two stage generation pipeline. Thus we choose only one of them for comparison, considering the effort required to retrain them on ScanNet v2.
>
> 4. **[Q1]** Scalability.
>
> Thanks for this insightful suggestion. Terra indeed relies on the amount and quality of 3D training data, just like any other deep learning models. The minimal requirement for 3D data of Terra is RGB and point cloud. One possible way to scale up Terra beyond current readily available 3D datasets might be a two stage strategy: large data pretraining and high-quality data finetuning, which is the same as typical video generation models. We could collect a large set of RGB images and use off-the-shelf reconstruction method to estimate the depth and poses, and then use this data to train Terra. Based on the pretrained version, we could further finetune it on some high-quality dedicated 3D data to improve the performance.
>
> 5. **[Q2]** Training time.
>
> We train P2G-VAE for 576 GPU hours and SPFlow for 1344 GPU hours.
>
> 6. **[Q3]** Generation range.
>
> We are sorry for the confusion. A single generation step covers 2.4m*2.4m, and the exploration can be done infinitely in theory. And the maximum range for exploration in our demonstration is around 7m*7m.
>
>  7. **[Q4]** Reconstruction setting.
>
> We are sorry for the confusion. We report the metrics for reconstruction on novel views. We will make modifications in the revised version.

---

### Official Review · Reviewer_jrCy · 2025-11-03

**Soundness:** 3
**Presentation:** 3
**Contribution:** 2
**Rating:** 4
**Confidence:** 3

**Summary:**

This paper introduces Terra, a framework for generating explorable 3D environments, where each environment is represented by point latents. The overall pipeline consists of two main components:
1. A Point-to-Gaussian VAE (P2G-VAE) based on PTv3, which encodes colored 3D points into a compact latent space.
2. A Sparse Point Flow Matching (SPFlow) model that learns the distribution of these point latents in the latent space.

The method is trained and evaluated on the ScanNet v2 dataset.
- For reconstruction, Terra achieves better depth accuracy than PixelSplat, MVSplat, Prometheus, and Can3Tok, though it performs worse on the LPIPS metric.
- For unconditional/image-conditioned generation, its geometric quality surpasses Trellis and Prometheus, while its visual quality is higher than Trellis but below Prometheus.

Ablation studies show that:
- Robust Position Perturbation reduces reconstruction quality in P2G-VAE but significantly enhances generative capability.
- Adaptive Upsampling and Refinement and Explicit Color Supervision both improve reconstruction and generation performance.
- Distance-Aware Trajectory Smoothing plays a key role in stabilizing training for generation tasks.

**Strengths:**

1. The proposed Distance-Aware Trajectory Smoothing is novel and demonstrates clear effectiveness in the context of sparse point flow matching models.

**Weaknesses:**

1. The term “world model” is conceptually broad. Using it as the paper’s main title may be misleading, as the method focuses more narrowly on 3D world generation and exploration, primarily within indoor scenes.
Moreover, most compared methods do not explicitly position themselves as world models — e.g., Prometheus (text-to-3D generation), Can3Tok (3D scene-level generation), and PixelSplat/MVSplat (3D reconstruction).

**Questions:**

1. What is the motivation for removing all residual connections in PTv3?
2. What exactly is the ground-truth distribution $\mathbf{P}$ of point latents, and how is it sampled? During inference, are the positions in $\mathbf{P}$ also randomly sampled from Gaussian noise?
3. How do the authors position Terra relative to recent approaches such as WorldMem [Xiao et al., 2025], VMem [Li et al., 2025], and Voyager [Huang et al., 2025]? From a visual standpoint, Terra’s generated results appear somewhat blurry, incomplete, or low-resolution compared to these models. While it remains an open question what the ideal representation for world models should be (e.g., 3D Gaussian Splatting, video-based, or otherwise), it would strengthen the paper if the authors clarified why comparisons to these methods were omitted and articulated Terra’s distinct advantages or future potential.

Things to improve the paper that did not impact the score:
- Please report the GPU hours required to train P2G-VAE and SPFlow.
- Table 1 appears far from its first citation — consider adjusting its placement for readability.
- Consider adding a section on the use of large language models (LLMs).

---

> ### Author Response · Authors · 2025-11-14
>
> We thank the reviewer for the constructive comments. We provide our answers below.
> 1. **[W1]** Clarification on world model.
>
> Thanks for the thoughtful suggestion. Indeed, there is no one official definition for world models currently. We think the ultimate goal of a world model is constructing a digital twin of the real world which faithfully represents the spatial and temporal evolution of the real world. In this paper, Terra focuses on the representation choice of world models and validates it on learning the spatial distribution of real scenes. We conduct our experiments on indoor scenes to focus on representation comparisons and also because of computation constraints. We will include more discussions about world models in the revised version.
>
> 2. **[Q1]** Residual connections in PTv3.
>
> We are sorry for the confusion. We remove the residual connections between the encoder and decoder of PTv3 to convert it into a VAE. When using the decoder of the VAE, we do not have the input to the encoder, so there should not be residual connections.
>
> 3. **[Q2]** Ground-truth distribution of point latents.
>
> We are sorry for the confusion. The GT of point latents is obtained by passing a colored point cloud through the P2G VAE encoder. Thus the GT distribution of point latents is determined by the distribution of colored point cloud and the deterministic VAE encoder. During training, we randomly sample a colored point cloud from the dataset to sample point latents. During inference, we randomly sample the positions of point latents from Gaussian noise just the same as standard diffusion.
>
> 4. **[Q3]** Additional comparisons.
>
> Thanks for the nice suggestion. In the paper, we compare Terra with world models based on 2D and 2.5D representations. We think WorldMem, VMem, Voyager can be categorized into 2D, 2.5D, 2.5D world models, respectively, which could suffer from multi-view inconsistency due to ambiguous cross-view pixel correspondences. Compared with them, Terra uses native 3D representation as the basis for world modeling, which achieves 3D consistency with rasterization without bells and whistles. Also, Terra supports efficient multiple renderings from different trajectories without rerunning the generation pipeline. We think Terra is promising to achieve universal world modeling since the real world is inherently three-dimensional and it is better to learn its distribution in its most natural form, although we indeed still have a long way to go. We will provide more discussions in the revised version.
>
> 5. **[Q4]** GPU hours.
>
> We train P2G-VAE for 576 GPU hours and SPFlow for 1344 GPU hours.
>
> 6. **[Q5 & Q6]** Readability and LLM.
>
> Thanks for the suggestion. We will make modifications in the revised version.

---

### Official Review · Reviewer_mSMJ · 2025-11-03

**Soundness:** 1
**Presentation:** 2
**Contribution:** 2
**Rating:** 2
**Confidence:** 5

**Summary:**

The paper introduces a method that, given an input colored point cloud, generates shapes in a latent space and allows for progressive exploration. The method uses a Point-to-Gaussian VAE to compress 3D inputs into sparse point latents and decodes them into 3D Gaussian primitives for rendering. It then uses a sparse point flow matching model to jointly denoise point positions and features for generative modeling.

**Strengths:**

The paper attempt to tackle the reconstruction problem from a native 3D generation perspective. The idea makes sense.

**Weaknesses:**

1) The method utilizes an input point cloud from fused multi-view depth sensors, which should provide high-quality shapes and textures. However, the generated results appear to make both the shapes and texture blurrier.


2) The comparison with Trellis is unfair. A more appropriate baseline would be to add the point cloud condition to Trellis, for instance, by voxelizing the point cloud to serve as the sparse grid for trellis's structured latent.

3) It seems that the paper is regenerating things that is already available from the input. In Fig. 1. What portion of the generated scene is not present in the input?   It is recommend to visualize the difference between input point cloud and the generated one.


4) The paper is missing comparisons with important RGB-D reconstruction baselines, such as classic depth map fusion methods (e.g., BundleFusion [1], ElasticFusion [2]) or methods based on neural fields [3]. The reconstruction results reported in this paper appear to be much worse than those achieved by the aforementioned baselines.

[1] BundleFusion: Real-time Globally Consistent 3D Reconstruction using Online Surface Re-integration

[2] ElasticFusion: Real-time dense visual SLAM system

[3] Neural RGB-D Surface Reconstruction

**Questions:**

- Is this method trained on random 3D crops?
- Does this method complete occluded geometry?
- Line 462. Can this method explorable unseen geometry in the input point clouds?

---

> ### Author Response · Authors · 2025-11-14
>
> We thank the reviewer for the constructive comments. We provide our answers below.
> 1. **[W1]** Blurred shapes and texture.
>
> We thank the reviewer for pointing this out. Indeed, the generated samples of Terra still lack intricate geometry and delicate texture. We attribute this limitation to the suboptimal quality of the 3D data. The fused point cloud from ScanNet v2 often contains noisy points, random holes and occluded parts due to sensor failure and limited viewpoints, which confuses Terra in learning the 3D geometry distribution. In addition, the images from ScanNet v2 suffer from serious motion blur which makes it hard to learn a high-fidelity texture. Despite this, we are actively collecting more high-quality 3D data to train our model.
>
> 2. **[W2]** Comparison with Trellis.
>
> Thanks for the suggestion. The original pipeline of Trellis directly takes in an image as condition, and thus we follow the official instructions to conduct the image-conditioned generation experiments. Despite this, we will try to use a partial voxelized point cloud as condition to Trellis in the revised version.
>
> 3. **[W3]** Misunderstanding about Figure 1.
>
> We are sorry for the confusion. The left part of Figure 1 demonstrates the exploration application of Terra which is purely generated without image or point cloud input during exploration. The bird's eye views and simulated camera views are both renderings of the generated 3D room. The right part of Figure 1 should actually be separated into two reconstruction and generation parts. The generative world modeling module operates as an individual separate stage, outputting point latents which are then transformed into Gaussian primitives with the P2G decoder. We will modify Figure 1 in the revised version.
>
> 4. **[W4]** Comparison with RGB-D reconstruction baselines.
>
> Thanks for the suggestion. The reconstruction module of Terra (P2G-VAE) is a variational autoencoder, which compresses the input 3D data into compact representations suitable for generative modeling. Thus the compression part is actually highlighted in the VAE paradigm. However, the RGB-D reconstruction baselines such as BundleFusion and NeRFs are dedicated reconstruction models with a quite different purpose and application scenario.
>
> 5. **[Q1]** Training details.
>
> We are sorry for the confusion. The reconstruction training is based on the full scene, while the generation training is based on random 2.4m*2.4m crops of the full scene.
>
> 6. **[Q2]** Occluded geometry.
>
> Yes, Terra is able to complete occluded geometry thanks to the masked conditional generation training, which would mask out random crops of the sample and train the model to complete them. Examples can be found in Figure 1 and 7 where Terra generates occluded structures beyond a single surface.
>
> 7. **[Q3]** Explore unseen geometry.
>
> Yes, Terra is able to explore unseen geometry as illustrated in Figure 1 and 7. Actually, when we use Terra for exploration, we do not use input point cloud. The renderings from Figure 1 and 7 are generated by Terra without point cloud guidance.

---

### Official Review · Reviewer_fH8q · 2025-11-03

**Soundness:** 3
**Presentation:** 3
**Contribution:** 3
**Rating:** 4
**Confidence:** 2

**Summary:**

The paper addresses the fundamental limitation of existing world models that rely on pixel-aligned representations. The authors introduce Terra, a native 3D world model, that represents and generates explorable environments using an intrinsic 3D latent space through two key technical innovations: a Point-to-Gaussian Variational Autoencoder (P2G-VAE) that encodes 3D inputs into latent point representations and decodes them as 3D Gaussian primitives to jointly model geometry and appearance, and a Sparse Point Flow Matching Network (SPFlow) that generates latent point representations by simultaneously denoising positions and features of point latents.

**Strengths:**

1. The paper is well-structured and easy to follow
2. Point-to-Gaussian Variational Autoencoder (P2G-VAE) effectively reduces redundancy in 3D input data while creating a compact latent space that jointly models both geometry and appearance through 3D Gaussian primitives, making it highly efficient for generative modeling.
3. Flexible rendering from any arbitrary viewpoint with only a single generation process
4. Progressive training strategy with three well-designed stages (reconstruction, unconditional pretraining, masked conditional generation)

**Weaknesses:**

1. No inference / training time comparison
2. No memory usage analysis
3. I believe that performance relates more to the method timing and suggest to use terms "Reconstruction Accuracy" and "Generation Accuracy" in tables 1 and 2.
4. I suggest a couple of high resolution renders in appendix or videos in supplementary materials to evaluate a visual quality of Terra


Overall, I'd be glad to increase the score if the authors address the above issues

**Questions:**

see weaknesses section

---

> ### Author Response · Authors · 2025-11-14
>
> We appreciate the constructive comments from the reviewer. We give our answers below.
> 1. **[W1 & W2 & W3]** Inference / training time and memory consumption.
>
> Thanks for the suggestion. The VAE and diffusion networks of Terra are based on standard PTv3 and OACNN, respectively, and thus the runtime statistics should be quite optimized and at least on par with other methods. Here we provide rough values and we will report accurate ones in the revised version for a fair performance comparison.
> |                              | Training Time  |  Inference Time  | Inference Memory |
> | :---                        |    :----:             |          :----:          | :----:                      |
> |Reconstruction      |  ~5 s/iter         |           TBD          |    <8GB                 |
> | Generation           |  ~6 s/iter          | 8s (50 steps)      |    <10GB               |
>
> 2. **[W4]** High resolution renders and video demonstrations.
>
> Thanks for the advice. We will provide a comprehensive video demonstration for the reconstructed and generated 3D assets in the revised version.

---

### Author Response · Authors · 2025-11-14

We thank the reviewers and ACs for their efforts and constructive comments. We have tried our best to answer the questions on a very tight schedule. We will continue to improve our work.

---

### Note · Authors · 2025-11-14

I have read and agree with the venue's withdrawal policy on behalf of myself and my co-authors.